# Relation of Maternal Pre-Pregnancy Factors and Childhood Asthma: A Cross-Sectional Survey in Pre-School Children Aged 2–5 Years Old

**DOI:** 10.3390/medicina59010179

**Published:** 2023-01-16

**Authors:** Dimitrios Papandreou, Eleni Pavlidou, Stefanos Tyrovolas, Maria Mantzorou, Eleni Andreou, Evmorfia Psara, Georgios Antasouras, Georgios K. Vasios, Efthymios Poulios, Constantinos Giaginis

**Affiliations:** 1Department of Health Sciences, College of Natural and Health Sciences, Zayed University, Abu Dhabi P.O. Box 144534, United Arab Emirates; 2Department of Food Science and Nutrition, School of Environment, University of the Aegean, 81400 Myrina, Lemnos, Greece; 3Department of Nursing, The Hong Kong Polytechnic University, Kowloon 999077, Hong Kong; 4Research, Innovation and Teaching Unit, Parc Sanitari Sant Joan de Déu, 08830 Sant Boi de Llobregat, Spain; 5Instituto de Salud Carlos III, Centro de Investigación Biomédica en Red de Salud Mental, CIBERSAM, 28029 Madrid, Spain; 6Department of Life and Health Sciences, University of Nicosia, 46 Makedonitissas Ave, P.O. Box 24005, Nicosia 1700, Cyprus

**Keywords:** childhood asthma, maternal risk factors, pre-school age, maternal obesity, caesarean section

## Abstract

*Background and Objectives*: Asthma constitutes a constant, prolonged, inflammation-related pulmonary disorder in childhood with serious public health concerns. Several maternal risk factors can enhance the prevalence of its development in this stage of life; however, the currently available data remain contradictory and/or inconsistent. We aim to evaluate the potential impacts of mothers’ sociodemographic, anthropometric and prenatal and perinatal factors on the prevalence of developing asthma in pre-school children. *Materials and Methods*: This is a retrospective cross-sectional survey, which includes 5133 women and their matched pre-school children. Childhood asthma was diagnosed using validated questionnaires. Statistical analysis was accomplished to evaluate whether maternal sociodemographic, anthropometric and prenatal and perinatal factors can increase the probability of childhood asthma in pre-school age. *Results*: A prevalence of 4.5% of childhood asthma was recorded in pre-school age. Maternal age and pre-pregnancy overweight and obesity, caesarean section, gestational diabetes and hypertension and not breastfeeding were associated with childhood asthma after adjustment for multiple confounding factors. *Conclusion*: Our research showed that several maternal factors increase the prevalence of childhood asthma in pre-school age. Suitable and effective health policies and strategies should be taken into account to confront the predominant maternal factors that increase its prevalence in pre-school age.

## 1. Introduction

Asthma constitutes a prolonged, inflammation-related pulmonary non-communicable disorder in children, with asthma symptom 12-month prevalence of approximately 11–14% in children and adolescents [1]. Notably, a prevalence of 8.4% of the population in the United States has been diagnosed with asthma and an incidence of 4.3% was recorded worldwide; these frequencies are continuously increasing [2]. Concerning the mechanism of asthma, persistent airway inflammation and hyper-responsiveness may take place [3]. Regarding its symptomatology, dyspnea, wheezing and coughing, as well as varying restrictions of expiratory airflow, occur [3]. The most important causes for developing asthma deal with the pollution of the environment, obesity, pet allergen exposure and several prenatal risk factors [4]. Asthma is currently considered one of the leading causes of childhood respiratory illnesses and is expected to continue to increase the disease burden in the future, rendering it an important public health objective [5]. Importantly, asthma is related to decreased lung activity from the beginning of infancy or at a prenatal stage, persisting in childhood and adulthood and predisposing to early or more serious chronic obstructive pulmonary disease (COPD) [6]. Currently, evidence is gradually increasing concerning the potential relationship between maternal prenatal and perinatal risk factors and childhood asthma. Several studies have suggested that childhood asthma diagnosis may be associated with maternal age, maternal overweight and obesity, maternal socioeconomic status and smoking habits, gestational weight gain and mode of delivery [7,8,9]. Mothers‘ overweight/obesity before pregnancy has been shown to be one of the most crucial factors for maternal and neonatal negative health effects, including childhood asthma [10]. However, most studies showed inconsistent and/or inconclusive findings, especially those concerning the effect of maternal obesity and weight gain during pregnancy on developing childhood asthma [11,12]. Several pieces of evidence support the association of delivery mode with the prevalence of bronchial asthma; however, the existing findings remain contradictory [8,13,14,15]. In addition, gestational diabetes is associated with the enhanced prevalence of asthma and wheezing; however, further studies are strongly recommended to clarify the exact factors affecting this relationship [16]. Reduced childbirth weight and early birth are considered among the risk factors related to childhood asthma; however, data from meta-analyses have found inconsistent findings [17]. Mothers’ smoking and third-hand smoke has also been found to influence its prevalence and strictness [18]. Some studies also suggested that exclusive breastfeeding may prevent childhood asthma, whereas other studies did not confirm a positive association [19,20,21]. Recently, a new study by Wilson et al. [21] showed that the longer the period of exclusive breastfeeding, the lower the odds of developing asthma later on. This is possibly due to the support of lung growth and enhancement of lung function. Since data in Greece are limited, these findings may have important clinical and public health implications for the improvement of the Greek health system.

In this aspect, the present retrospective study aims to evaluate the potential impacts of maternal sociodemographic, anthropometric and prenatal and perinatal factors in the prevalence of developing asthma in children aged 2–5 years.

## 2. Materials and Methods

### 2.1. Subjects

In the present retrospective study, 7038 pre-school children aged 2–5 years and their matched mothers were enrolled from nine geographically diverse Greek regions, namely Athens, Thessaloniki, Larisa, Patra, Alexandroupolis, Kalamata, Ioannina, Crete and the North Aegean. Recruitment to the study was between the period May 2016 and September 2020. A flow chart of study enrolment is presented in Figure 1. The inclusion criteria for the initial enrollment were children aged 2–5 years whose mothers had a singleton birth in the prior 2–5 years. All participants’ information was confidential, and all participating children were disease-free except for possible development of asthma or diabetes melittus I during the postpartum period. Among the 7038 initially enrolled children and their matched mothers, 845 were excluded from the study due to missing or incomplete data. Among the remaining 6193 children and matched mothers, 1060 of the participating children were then excluded from the study due to any history of disease such as neurodevelopment disorders (e.g., autism spectrum disorder, attention deficit hyperactivity disorder, mental retardation, motor disorder), diabetes mellitus II, hypertension, hyperinsuinemia, anemia, hyperlipidemia, cancer, etc. The above exclusion was applied in order to assess whether maternal obesity affected the prevalence of childhood obesity, independently of other childhood diseases which may increase the risk for childhood obesity, such as metabolic disorders or hypertension, and for which a considerably higher sample size would be required. The final response rate was equal to 70.3%. The history of the potential disease of the excluded children was reported by their mothers in the questionnaires provided. A total of 5133 children and their matched mothers were included in the final analysis after the above inclusion and exclusion criteria were applied. The mothers of the children were informed of the purpose of the study and signed a consent form. All participating mothers were kept updated concerning the aim of the study and endorsed the consent form. The study was endorsed by the Institutional Ethics Committee of the University of the Aegean (Ethics approval code: no 12/14.5.2016) in accordance with the World Health Organization criteria (52nd WMA General Assembly, Edinburgh, Scotland, 2000). The endorsed children had no other disorders except for a possible history of asthma. The mothers were selected randomly during their visits to their personal gynecologists or in public or private health units, as well as visits to schools, kindergartens and playgrounds.

### 2.2. Study Design

At the moment of study (2–5 years postpartum), semi-quantitative questionnaires were utilized to evaluate social and demographic parameters, lifestyle and prenatal factors of mothers and their children [22]. The questionnaires were completed by the mothers of the children. The weight of the enrolled women at the beginning of gestation and just prior to birth was obtained from their private medical files. The mothers’ weight was measured for the duration of their follow-up at public or private hospitals. Mothers’ weight gain during gestation was computed from the measured weight just prior to birth minus the measured weight at the beginning of gestation. Mothers’ anthropometric parameters (weight and height) at the moment of study (2–5 years after delivery) were also determined by qualified nutritionists and physicians. The same electronic scale was used to measure body weight, and a portable stadiometer was utilized to measure height. Body mass index (BMI) was determined as (weight (kg)/height (m)^2^).

Additionally, childhood diabetes melittus type 1 and preterm birth (<37th week) were recorded from the mothers of the study children in the questionnaire provided. Mothers’ answers concerning preterm birth were further cross-checked by their gynecologists’ or hospitals’ medical files for more precise records for the exact week of preterm birth to be obtained; however, we observed that there were several missing data concerning the exact week of preterm birth and several of them did not agree with the mothers’ answers. Thus, preterm birth was treated as a binary outcome before and after the 37th week of pregnancy.

Smoking habits, years of education and financial level were self-reported by the enrolled mothers. Smoking status was classified as smokers (systematic traditional tobacco smoking of at least 5 cigarettes every day) and never smoking (non-smoking at all). Financial level was categorized based on their yearly family salary: EUR 0 ≤ 5000, EUR 1 ≤ 10,000, EUR 2 ≤ 15,000, EUR 3 ≤ 20,000, EUR 4 ≤ 25,000 and EUR 5 ≥ 30,000. Economic level was additionally categorized as low for a yearly salary of EUR ≤ 10,000, medium for a yearly salary of EUR > 10,000 and EUR ≤ 20,000 and high for a yearly salary of EUR > 20,000. Breastfeeding practices, especially breastfeeding for a minimum of a period of four months, were also investigated. Additionally, doctors’ diagnoses of gestational diabetes and gestational hypertension were recovered by women’s private medical files. The mode of delivery (vaginal or caesarean section), as well as a history of preterm birth (<37th week), were self-reported by the enrolled women.

Childhood asthma was diagnosed by specialized physicians according to the International Study of Asthma and Allergies in Children and information regarding asthma-specific therapy and health care usage [23,24].

Maternal reports of child wheezing from questionnaires derived from the ISAAC were used. We considered wheezing to be present if the maternal parent answered “yes” to the question: “In the past 12 months, has your child ever had wheezing (or whistling in the chest)?” [24]. In addition, asthma was further defined by a minimum of three events of wheezing in conjunction with treatment with glucocorticosteroids, signs of suspected hyper-reactivity without concurrent upper respiratory infection and if their child’s sleep had been troubled due to wheezing by a minimum of one night/week in the last one month [24].

Descriptive guidelines were provided to the enrolled women by qualified nutritionists and physicians concerning the questionnaires’ accomplishment, and a comprehensive demonstration of the questions to enable reliable responses was facilitated.

### 2.3. Statistical Analysis

Statistical analysis was accomplished by the Student’s *t*-test and one-way ANOVA for continuous variables, following normal distribution. Normality distribution was evaluated by the Kolmogorov–Smirnov test. The chi-square test was utilized for categorical variables. The Mann–Whitney non-parametric test was applied for continuous variables between 2 groups that did not follow normal distribution. The Kruskal–Wallis non-parametric test was performed for continuous variables between 3 or more groups. The quantitative variables, following normal distribution, are stated as the mean value ± standard deviation (SD). The continuous quantitative variables, which do not follow normal distribution, are provided as median values (interquartile range, IQR) and the qualitative variables are presented as absolute or relative frequencies. Multivariate logistic regression analysis was applied to evaluate whether childhood asthma is related to mothers’ sociodemographic, anthropometric and lifestyle features, as well as mothers’ prenatal and perinatal factors by adjusting for multiple confounding factors, e.g., maternal age, pre-pregnancy overweight or obesity, educational and economic status, smoking habits, preterm birth, mode of delivery, breastfeeding practices, gestational weight gain and gestational diabetes and hypertension. The results of multiple regressions are presented as odds ratios (OR) at a 95% confidence interval (CI). Differences were considered statistically significant if *p* < 0.05. The statistical analysis of the survey data was accomplished by Statistica 10.0 software, Europe (Informer Technologies, Inc., Hamburg, Germany).

## 3. Results

### 3.1. Maternal Sociodemographic Anthropometric and Lifestyle Characteristics and Perinatal Outcomes of the Study Population

At the time of the study (2–5 years postpartum), 5133 women were enrolled with a mean age of 37.54 ± 4.85 years. Regarding their ethnicity, 95.7% were Greek, and the remaining 4.3% were of other ethnicities. The mean educational years were 15.1 ± 2.2 years (range: 6–17 years). In total, 45.6% of the participant women reported low financial status, and 54.4% had medium or high economic levels compared to the medium. In addition, 25.6% of the mothers reported smoking habits before and after gestation. During gestation, none of the mothers smoked except for 2.1% that reported occasional smoking of 1–2 cigarettes during a period of 1–2 weeks.

The mean BMI pre-pregnancy was 22.7 ± 3.7 kg/m^2^ (range: 15.9–37.6 kg/m^2^). At the time of the study (2–5 years after delivery), their BMI was significantly higher at 23.7 ± 4.4 kg/m^2^ (range: 16.3–41.5 kg/m^2^, *p* < 0.0001). In particular, 17.5% of the enrolled mothers were affected by overweight, and 5.0% were affected by obesity before gestation. Overall, an incidence of 22.5% was characterized by overweight or obesity before gestation. At the time of the study (2–5 years postpartum), 21.0% of the women were affected by overweight, and 9.6% were affected by obesity, with an overall overweight/obesity proportion of 30.6%. Concerning breastfeeding practice, 50.3% of the enrolled women breastfed exclusively for at least four months (duration: 4.8 ± 1.9 months), and the remaining 49.7% did not follow exclusive breastfeeding for at least 4 months or did not breastfeed at all.

The mean body weight gain during pregnancy was 13.8 ± 6.1 kg (range 4.0–45.0 kg). An incidence of 30.2% for preterm birth (<37th week) was noted. A caesarean section was performed in 56.4 % of the study population, while among the rest, 43.6% delivered vaginally. Additionally, an incidence of 4.3% of the enrolled women had a history of gestational diabetes mellitus, and 4.2% of the enrolled mothers had a history of gestational hypertension.

### 3.2. Childhood Asthma in Association with Maternal Sociodemographic and Anthropometric Factors

Childhood asthma was recorded in 4.5% of the enrolled children. Childhood asthma was significantly more frequently observed in children delivered by older mothers than those born to younger mothers (Figure 2A, Table 1, 37.6 ± 4.8 vs. 36.6 ± 5.8 years, *p* = 0.0023). Greek children had a marginally higher prevalence to present asthma symptoms than children born by mothers of other nationalities (Table 1, *p* = 0.0475). Mothers whose children were diagnosed with asthma had significantly higher mean pre-pregnancy BMI than mothers whose children did not develop asthma at pre-school age (Figure 2B, Table 1, 23.4 ± 3.3 vs. 22.7 ± 3.7 kg/m^2^, *p* = 0.0048). Analyzing pre-pregnancy BMI status by categories, overweight and obese mothers had a significantly higher prevalence of childhood asthma compared to normal-weight mothers (Table 1, *p* < 0.0001). Childhood asthma was not associated with maternal educational level, economic status and smoking habits (Table 1, *p* > 0.05).

### 3.3. Childhood Asthma in Association with Maternal Perinatal Factors


Gestational weight gain was higher in mothers whose children were diagnosed with asthma symptoms than those without asthma symptoms (Figure 2C, Table 1, 14.8 ± 6.6 kg vs. 13.8 ± 6.3, *p* = 0.0147). In cross-tabulation, children born by caesarean were more frequently diagnosed with asthma than those delivered vaginally (Table 1, *p* = 0.0034). Asthma prevalence was higher in children whose mothers developed gestational diabetes (Table 1, *p* = 0.0003), as well as in those whose mothers presented pregnancy-induced hypertension (Table 1, *p* < 0.0001). Children that did not develop asthma symptoms more frequently breastfed exclusively for a minimum period of four months (Table 1, *p* = 0.0006). Childhood asthma was not associated with preterm birth (Table 1, *p* > 0.05). We further performed an analysis for a subgroup of mothers with data available (n = 2.195) who exclusively breastfed for at least 6 months without finding any significant difference compared to those breastfeeding for at least four months (data not shown). We also performed a subgroup analysis by dividing the enrolled children into two groups: children aged 2–3 years vs. children aged 4–5 years, and we did not find any significant difference that may be influenced by environmental factors (data not shown).

### 3.4. Multivariate Regression Analysis for Childhood Asthma

In the multivariate logistic regression analysis, childhood asthma was independently associated with maternal age, pre-pregnancy BMI status, exclusive breastfeeding, mode of delivery, gestational diabetes and gestational hypertension by adjusting for multiple confounding factors (Table 2, *p* < 0.05). Older mothers had a 32% higher prevalence of delivering children who developed asthma 2–5 years postpartum compared to younger mothers (Table 2, *p* = 0.0245). Mothers affected by overweight/obesity pre-pregnancy exhibited 87% higher odds of delivering children who presented asthma symptoms 2–5 years postpartum than those that were underweight and normal weight pre-pregnancy (Table 2, *p* = 0.0179). Mothers following exclusive breastfeeding for a minimum period of four months had two-fold lower odds of delivering children who developed asthma 2–5 years postpartum compared to mothers not following exclusive breastfeeding (Table 2, *p* = 0.0095).

Children delivered by caesarean section had an 89% higher incidence to present asthma symptoms 2–5 years postpartum than those born vaginally (Table 2, *p* = 0.0194). Mothers who were diagnosed with gestational diabetes had a 43% higher probability of delivering children who presented asthma symptoms 2–5 years postpartum than those who did not diagnose with gestational diabetes (Table 2, *p* = 0.0298). Moreover, mothers who developed pregnancy-induced hypertension had 14% higher odds of delivering children with asthma symptomatology 2–5 years postpartum than those who did not develop gestational hypertension (Table 2, *p* = 0.0130). Mothers’ ethnicity, education and financial status, smoking history and body weight gain during gestation, as well as preterm birth, did not exert any significant, independent effects on childhood asthma prevalence in multivariate analysis (Table 2, *p* > 0.05).

## 4. Discussion

Childhood asthma constitutes a severe public health concern which may be associated with multiple maternal risk factors; however, the existing available studies provide contradictory results for certain risk factors, while several seem inconclusive. In this aspect, the present retrospective study aimed to investigate the potential associations between childhood asthma and various maternal risk factors, such as sociodemographic, anthropometric and prenatal and perinatal factors in pre-school children in Greece.

Firstly, we found that older maternal age was related to a greater prevalence of childhood asthma, 2–5 years postpartum. In this aspect, some studies suggested that maternal age is positively related to childhood asthma, whereas others did not support any association [25,26]. In a recent large prospective cohort, children whose mothers were older and had a history of asthma had a higher prevalence of persistent asthma than those with younger mothers; however, this relationship did not remain significant in the case of mothers who were not previously diagnosed with asthma [27]. Unfortunately, we have no available data for maternal asthma history except for a subclass (n = 1294) of our study population in which a prevalence of 12.4% had developed asthma in the past. In this subclass, older mothers had a higher incidence of asthma concerning their children, independently of maternal asthma history (data not shown).

There is also evidence suggesting that maternal pre-pregnancy overweight/obesity may be considered a crucial cause of childhood asthma [10,26,28,29]. In accordance with the existing data, we found analyzing pre-pregnancy BMI status by categories that overweight and obese mothers had a significantly higher prevalence of childhood asthma compared to normal-weight mothers. In a meta-analysis of 22 observational studies, maternal pre-pregnancy overweight/obesity significantly increased the prevalence of children diagnosed with asthma [30]. Nevertheless, the majority of studies showed inconsistent and inconclusive findings due to the confounding effect of maternal asthma history [11,12]. The causal effect of mothers’ obesity on children’s asthma remains doubtful despite the numerous studies investigating these relations and which in parallel take into consideration both major confounding factors and possible mediators [10]. In addition, further analyses regarding the transcriptome of the placenta of normal weight and obese women showed that women with obesity had altered gene expression in the placenta, affecting the in utero conditions for the development of the embryo [31].

In our study population, pre-pregnancy maternal overweight and obesity were correlated with an 87% higher probability of childhood asthma. When performing the same analysis in the subclass of our study population with available data for maternal asthma history, we found a strong association in the subgroup of mothers with asthma history but only a marginal association in the subgroup of mothers without asthma history (data not shown).

A small number of surveys have currently assessed the impact of gestational weight gain on the prevalence of childhood asthma. More to the point, there is evidence that higher body weight gain during pregnancy than the recommended was generally not related to childhood asthma [29,30,32]. In contrast, some studies and a meta-analysis reported that lower or higher than the recommended maternal gestational weight gain rendered their children predisposed to a considerably increased likelihood of developing asthma in the first years of their life [33,34]. In our study population, we found that gestational weight gain was higher in mothers whose children developed asthma 2–5 years postpartum. Nevertheless, this relationship was not substantial by adjusting for potential confounding factors such as maternal pre-pregnancy overweight or obesity and other maternal factors.

Exclusive breastfeeding exerted a protective impact as opposed to the development of childhood asthma, whereas other studies did not confirm a positive association [19,20,21]. There have also been three additional meta-analyses investigating asthma and breastfeeding in the last 10 years [35,36,37]. Two of them supported evidence that enhanced duration of breastfeeding and even any breastfeeding may protect against childhood asthma [35,36]. In contrast, the other meta-analysis did not show a considerable relationship between breastfeeding and asthma [37]. However, the above three meta-analyses mainly focused on case–control/cross-sectional studies with minimal quality and presented considerable heterogeneity, while the most recent meta-analysis performed by Lodge et al. focused on studies up to 2014 [35]. In the most recent meta-analysis, including 42 cohorts and randomized clinical trials, children with longer duration or more exclusive breastfeeding showed a decreased probability of asthma [38]. Notably, additional stratified evaluation of diverse age groups indicated a decreased probability of asthma in the age groups 0–2 years and 3–6 years, whereas there was no considerable impact on children aged ≥7 years [38]. In accordance with this recent meta-analysis, we found that exclusive breastfeeding is linked to more than two-fold smaller odds of developing asthma in pre-school children. The mechanism behind the association between breastfeeding and asthma protection may be the role of the microbiome, which is positively affected by breastfeeding and affects the offspring’s immunity and long-term health [39,40,41,42]. Additionally, it is important to note the impact of breastfeeding on the maturation of the neonatal immune system [43] and the development of tolerance to common allergens [44].

Several observational studies support evidence that the mode of delivery may increase the probability of childhood asthma. However, such findings remain inconclusive [8,13,14,15,45]. In this aspect, a recent meta-analysis demonstrated a higher probability of asthma in children born by caesarean section, both planned and unplanned, than those born vaginally [46]. However, this evidence should be taken with caution due to the considerable heterogeneity of the findings of the included studies [46]. In accordance with this meta-analysis, we found that children born by caesarean section exhibited an 89% higher incidence of developing asthma at 2–5 years old than those delivered vaginally. Notably, this association remained independent by adjusting for several confounders such as mothers’ age and pre-pregnancy overweight/obesity, education and economic status, smoking habits, gestational weight gain, preterm birth and exclusive breastfeeding. Gestational diabetes is associated with several complications, including prematurity and foetal distress, as well as obesity and type 2 diabetes in childhood [47]. Several studies also suggest that gestational diabetes may also exert permanent impacts on child health by alterations that affect foetal progression, including the growth of the lung and immune system [48,49].

Moreover, epidemiological surveys widely support the relationship between mothers’ gestational diabetes and children’s pulmonary consequences, such as asthma [50]. In a more recent, large-scale prospective cohort, gestational diabetes was related to a higher probability of childhood asthma approximately four years after delivery [16]. In line with the above, we found that gestational diabetes was related to a 43% higher probability of developing childhood asthma, and this evidence was vigorous after adjusting for multiple confounders.

Studies evaluating the relationship between maternal gestational hypertension and childhood asthma are currently scarce [51,52]. In fact, newborns from mothers with hypertension prior to or throughout pregnancy experienced a decrease in respiratory activity comparable to those subjected to mothers’ smoking during pregnancy. This is considered a well-recognized risk factor for reduced lung activity, early wheezing and childhood asthma at the age of 6 years [52,53]. In support of this view, our study found a 14% higher likelihood of diagnosis of asthma in children whose mothers presented pregnancy-induced hypertension, independently of several confounding factors.

Some studies suggested that mothers’ smoking habits are considered to influence the incidence and seriousness of developing asthma in childhood [18,54,55]. However, a cause–effect impact cannot be established as far as the available evidence is concerned. On the other hand, other studies did not support any difference concerning maternal smoking or not and the prevalence of childhood asthma [26]. Thus, prospective studies are required for conclusive results to be obtained. In this aspect, in our cross-sectional study, we did not find any causal effect of mothers’ smoking on the prevalence of asthma in pre-school children.

At least one study suggested that preterm birth can increase the prevalence of childhood asthma [56]. However, it was very early preterm birth, approximately 23–27 weeks of pregnancy, and not later preterm birth, which was linked to a higher probability of asthma, mainly in early adulthood [57]. In this aspect, our findings did not support any relationship between preterm birth and childhood asthma, which may be ascribed to the fact that we did not have extreme preterm births in our study population, as well as that our sample focused on childhood and not adulthood. Since the weeks of prematurity were between 35–37, we further run an analysis separately for 35, 36 and 37 weeks; however, we found no statistically significant difference between these weeks.

It should be noted that there are certain limitations to our study. Childhood asthma was diagnosed by specialized physicians using validated questionnaires from self-reports by mothers; thus, misclassification could not be excluded. In addition, it should be mentioned that, in young children, the diagnosis of asthma is thought to be less reliable because of its clinical variability in the first years of life [58]. Moreover, BMI is considered a crude measure to distinguish mothers’ who are overweight or obese. Nevertheless, direct methods assessing body fat quantity and distribution are required to expand and verify our results.

Furthermore, recall bias was inherent in our retrospective study since several potential risk factors were self-reported by mothers. Thus, no conclusions regarding causality can be proposed due to the present design of our survey despite its nationally representative nature. Additionally, in spite of a comprehensive attempt at confounders’ adjustment, we recognize the possibility of measureless confounders. In particular, several important factors associated with asthma in pre-school children could be considered in future studies, such as mothers’ age, exposure to secondhand smoking, past family asthma or respiratory infection prior to the age of two years old. Nevertheless, our study strength is the relatively large and representative study population since it included children from nine geographically diverse areas of Greece, including urban, rural and island regions.

## 5. Conclusions

This cross-sectional study supported evidence that there are several maternal prenatal and perinatal factors which may enhance the prevalence of childhood asthma in pre-school children. Maternal age, pre-pregnancy overweight/obesity, caesarean section, gestational diabetes and hypertension, as well as not exclusively breastfeeding, were significantly related to the elevated probability of childhood asthma in pre-school age. Regarding future research, large-scale and well-designed clinical studies are strongly recommended to confirm the present findings. Suitable and effective health policies and strategies should be applied to confront the predominant maternal risk factors that enhance the incidence of childhood asthma.

## Figures and Tables

**Figure 1 medicina-59-00179-f001:**
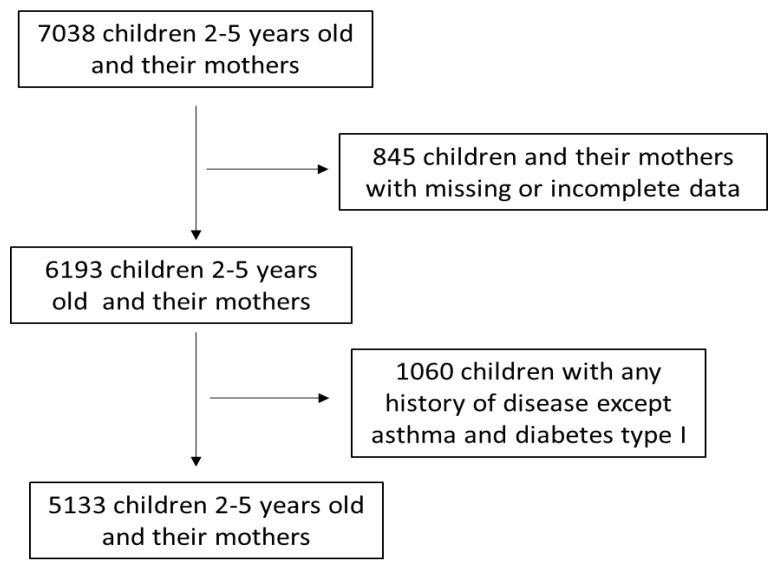
A flow chart of study enrolment.

**Figure 2 medicina-59-00179-f002:**
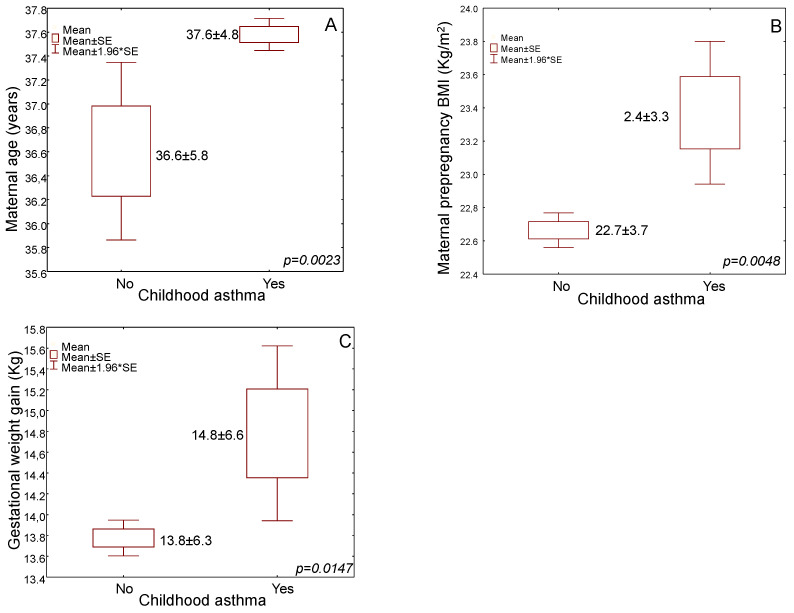
Box-whisker box plots of childhood asthma in association with (**A**) mothers’ age, (**B**) mothers’ BMI before gestation and (**C**) body weight gain during gestation.

**Table 1 medicina-59-00179-t001:** Associations of childhood asthma with maternal sociodemographic, anthropometric and prenatal and perinatal factors.

Parameters (*n* = 5133)	Childhood Asthma		
	No (95.5%)	Yes (4.5%)	*p*-Value
Age (years)	36.6 ± 5.8	37.6 ± 4.8	*p* = 0.0023
Nationality (n, %)			*p* = 0.0475
Greek	4684 (95.6)	229 (98.3)	
Other	216 (4.5)	4 (1.7)	
Pre-pregnancy BMI (kg/m^2^)	22.7 ± 3.7	23.4 ± 3.3	*p* = 0.0048
Pre-pregnancy BMI status (n, %)			*p* < 0.0001
Normal weight	3864 (78.9)	117 (50.2)	
Overweight	814 (16.6)	84 (36.1)	
The bold iObese	222 (4.5)	32 (13.7)	
Education (years ± SD)	15.1 ± 2.3	15.1 ± 2.0	*p* = 0.9505
Economic status (n, %)			*p* = 0.4309
Low	1356 (45.8)	58 (42.3)	
Medium or high	1607 (54.2)	79 (57.7)	
Smoking habits (n, %)			*p* = 0.0921
No smokers	3617 (73.8)	201 (86.3)	
Smokers	1283 (26.2)	32 (13.7)	
Gestational weight gain (kg)	13.8 ± 6.3	14.8 ± 6.6	*p* = 0.0147
Exclusive breastfeeding (n, %)			*p* = 0.0006
No	2441 (49.8)	143 (61.4)	
Yes	2459 (50.2)	90 (38.6)	
Preterm birth (<37th week, n, %)			*p* = 0.6947
No	3424 (69.9)	160 (68.7)	
Yes	1476 (30.1)	73 (31.3)	
Mode of delivery (n, %)			*p* = 0.0034
Vaginal	2160 (44.1)	80 (34.3)	
Caesarean section	2740 (55.9)	153 (65.7)	
Gestational diabetes (n, %)			*p* = 0.0003
No	4700 (95.9)	212 (91.0)	
Yes	200 (4.1)	21 (9.0)	
Pregnancy-induced hypertension (n, %)			*p* < 0000.1
No	4737 (96.7)	183 (78.5)	
Yes	163 (3.3)	50 (21.5)	

**Table 2 medicina-59-00179-t002:** Multivariate logistic regression analysis for childhood asthma adjusted for potential confounders.

Parameters	Childhood Asthma	
	HR ^a^ (95% CI ^b^)	*p*-Value
Age (Below/Over mean value)	1.32 (0.83–1.94)	*p* = 0.0245
Nationality (Greek/Other nationality)	0.98 (0.20–1.89)	*p* = 0.4501
Pre-pregnancy BMI status		*p* = 0.0002
Normal weight	1.0	
Overweight	1.63 (1.12–2.01)	
Obese	1.97 (1.65–2.34)	
Education (Below/Over mean value)	1.20 (0.32–2.19)	*p* = 0.2402
Economic status (Low/Medium or high)	0.75 (0.13–1.59)	*p* = 0.5611
Smoking habits (No/Yes)	1.53 (0.73–2.38)	*p* = 0.2749
Gestational weight gain (Below/Over mean value)	1.62 (0.92–2.46)	*p* = 0.0763
Exclusive breastfeeding (No/Yes)	2.25 (1.80–2.49)	*p* = 0.0095
Preterm birth (No/Yes)	1.29 (0.47–2.12)	*p* = 0.2873
Mode of delivery (Vaginal/Caesarean)	1.89 (1.41–2.30)	*p* = 0.0194
Gestational diabetes (No/Yes)	1.43 (0.94–2.01)	*p* = 0.0298
Pregnancy-induced hypertension (No/Yes)	1.14 (0.67–1.69)	*p* = 0.0130

^a^ Hazard Ratio: HR. ^b^ CI: Confidence Interval.

## Data Availability

Data are available upon reasonable request.

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
