# Peer review of "Relation of Maternal Pre-Pregnancy Factors and Childhood Asthma: A Cross-Sectional Survey in Pre-School Children Aged 2–5 Years Old"

_medicina, 2023, doi:10.3390/medicina59010179_

Round 1

Reviewer 1 Report

The proposed study aims to identify possible maternal risk factors for the development of asthma in early infancy.  As written in the introduction and discussion, the data reported in the cited literature about the risk factors considered are inconclusive, so the authors with their work would like to contribute to the identification and role of the risk factors considered in the development of asthma. 

The first point to clarify is the definition of asthma in the age considered (2-5 years old children). 

The ISAAC questionnaire cited above refers to an age range of a minimum of 6 years, and it is unclear whether for the definition of asthma, parents had to answer positively to all of the proposed questions. Furthermore, given that this information is reported by parents and a diagnosis was not made by a specialist, and considering the definition of asthma according to the GINA guidelines that states " asthma is a chronic airway inflammation. It is defined by the history of respiratory symptoms, such as wheeze, shortness of breath, chest tightness and cough, that vary over time and in intensity, together with variable expiratory airflow limitation," the term wheezing rather than asthma should be used in my opinion. 

Furthermore,I recommend directly citing the paper from which the questionnaire was taken and not the footnote 24 to which it refers.

Regarding the smoking factor in pregnancy, considering that most of the studies in the literature recognize it as a prenatal risk factor for the development of asthma, if you would like to include it in the study as a factor to be considered, I recommend analyzing several other variables such as the time exposure to smoke, the type of cigarettes smoked (traditional, heated tobacco or electronic cigarettes), the average number of cigarettes smoked during pregnancy.

Regarding prematurity, I recommend specifying the weeks of gestational age considered and possibly subdividing and re-analysing the subjects enrolled into two groups, according to the degree of prematurity. You could cite this paper on this topic in the discussion: doi:10.3390/children8100843.

Lastly in the title, please avoid abbreviation. I suggest also to be more general, because significance in the paper was identified in more than one risk factor not only in OW and OB and, on the contrary, it was not found in other factors besides breastfeeding.

Please check and correct english and typos.

Author Response

Dear reviewer,

Thank you very much for taking time to review our manuscript. We took into account your valuable comments, and we have revised our manuscript accordingly.

The proposed study aims to identify possible maternal risk factors for the development of asthma in early infancy.  As written in the introduction and discussion, the data reported in the cited literature about the risk factors considered are inconclusive, so the authors with their work would like to contribute to the identification and role of the risk factors considered in the development of asthma. 

The first point to clarify is the definition of asthma in the age considered (2-5 years old children). The ISAAC questionnaire cited above refers to an age range of a minimum of 6 years, and it is unclear whether for the definition of asthma, parents had to answer positively to all of the proposed questions. Furthermore, given that this information is reported by parents and a diagnosis was not made by a specialist, and considering the definition of asthma according to the GINA guidelines that states " asthma is a chronic airway inflammation. It is defined by the history of respiratory symptoms, such as wheeze, shortness of breath, chest tightness and cough, that vary over time and in intensity, together with variable expiratory airflow limitation," the term wheezing rather than asthma should be used in my opinion. 

Furthermore, I recommend directly citing the paper from which the questionnaire was taken and not the footnote 24 to which it refers.

Response: We have now added the correct citation for the questionnaire.

Regarding the smoking factor in pregnancy, considering that most of the studies in the literature recognize it as a prenatal risk factor for the development of asthma, if you would like to include it in the study as a factor to be considered, I recommend analyzing several other variables such as the time exposure to smoke, the type of cigarettes smoked (traditional, heated tobacco or electronic cigarettes), the average number of cigarettes smoked during pregnancy.

Response: We added the relative information both in methods and results sections

Regarding prematurity, I recommend specifying the weeks of gestational age considered and possibly subdividing and re-analysing the subjects enrolled into two groups, according to the degree of prematurity.

Response: The weeks of prematurity were between 35-37. We had already run an analysis separately for 35, 36 and 37 weeks  however we found no statistically significance difference between the weeks. We added a sentence in discussion to specify it.

You could cite this paper on this topic in the discussion: doi: 10.3390/children8100843.

Response: We have now added the ref.as suggested.

 Lastly in the title, please avoid abbreviation. I suggest also to be more general, because significance in the paper was identified in more than one risk factor not only in OW and OB and, on the contrary, it was not found in other factors besides breastfeeding.

 Response: We have now modified  the title as suggested.

Please also Correct typos.

Response: We have also corrected all typos mistakes

Reviewer 2 Report

To the authors,

The authors reported a cross-sectional cohort study to examine the association between maternal and perinatal factors with childhood asthma in Greek. The manuscript may need extensive English editing before further publication, and I have some suggestions for the study method and data presentation. 

Overall: 

Too many typos, and the citation format should be carefully checked before submitting your manuscript.

Please add some potential mechanisms to explain the relationship between maternal obesity, breastfeeding duration, and childhood asthma development. 

Introduction

Page 1, Line 40. “339.000 cases worldwide.”Please confirm the number of asthma children numbers. 

Page 1, Line 33. Citation 3

Page 2, Line 48. Considered currently considered? Please check it. 

Page 2, Line 50 Importnantly? Typos? 

Page 2, Line 53-57. Less correlated with childhood asthma for the discussion of asthma-COPD overlap syndrome. Please remove this part. You could add some description about how childhood asthma can lead to the severity of the likelihood of long-term pulmonary disease in adults

Page 2, Line 71. Citation 17

Page 2, Line 74-76. Please consider adding more discussion and explaining why more breastfeeding duration may lower the asthma risk. 

Methods

Page 2, Line 90-92. Please clarify why you have to use randomization in your study.

Page 3, Line 115 recovered ? Do you mean recorded?

Page 3, Line 120 citation 24

Page 3, Line 123 citation 24

Please consider to use “normal weight,” “overweight,” and “obese” to compare the asthma outcomes instead of BMI values only. You may compare the childhood asthma risk by categorizing maternal prepregnancy BMI into different categories to see if the association is still significant. BMI categories might correlate more to clinical practice than BMI values. 

Results

Page 4, Line 173.  56.4% + 43.7 % is not equal to 100%. Please revise the percentage numbers. 

Page 5 Figure 1. Please revise the resolution of your figure. It needs to be clarified for reading.

Table 1. Please list the BMI information as categories (overweight n (%), obese n (%)…). 

Please explain the number of preterm birth were up to 30 %. Why? 

Page 6, Line 195 typos. botn?

Page 7, table 2. Better to compare the risk by categorized BMI groups. 

Discussion 

Page 9, Line 314. Citation [26-45]? Please check it. 

Page 9, Line 328 citation 49

Please add more discussion about the potential mechanisms for maternal obesity, breastfeeding, and asthma development. 

Consider doing a subgroup analysis to know if a longer duration of breastfeeding could provide better protection effects of childhood asthma. For example, please provide the information to see if children with exclusive breastfeeding duration for six months may have a lower asthma risk than children without breastfeeding. 

Since asthma may be related to environmental factors, such as air pollution or obesity, please consider doing a subgroup analysis (younger vs. older age children; 2-3 years vs. 4-5 years) to see if the protective effect for asthma development may diminish or persist in older children. 

Author Response

Dear reviewer,

Thank you very much for taking time to review our manuscript. We took into account your valuable comments, and we have revised our manuscript accordingly.

Page 1, Line 40. 339.000 cases worldwide.”Please confirm the number of asthma children numbers. 

Response: We have revised the section to include that Asthma symptom 12-month prevalence is about 11-14% in children and adolescents.

Page 1, Line 33. Citation 3

Response: We have corrected the citation

Page 2, Line 48. Considered currently considered? Please check it. 

Page 2, Line 50 Importnantly? Typos? 

Response: We have corrected the typos in the manuscript

Page 2, Line 53-57. Less correlated with childhood asthma for the discussion of asthma-COPD overlap syndrome. Please remove this part. You could add some description about how childhood asthma can lead to the severity of the likelihood of long-term pulmonary disease in adults

Response: We have now removed this part

Page 2, Line 71. Citation 17

Response: We have corrected the citation

Page 2, Line 74-76. Please consider adding more discussion and explaining why more breastfeeding duration may lower the asthma risk. 

Response: We have now added a statement and a ref.

Methods

Page 2, Line 90-92. Please clarify why you have to use randomization in your study.

Response:This part was mistakenly added to the manuscript. We have now deleted the sentence.

Page 3, Line 115 recovered ? Do you mean recorded?

Response: We have corrected the typos in the manuscript

Page 3, Line 120 citation 24

Page 3, Line 123 citation 24

Response: We have corrected the citation

Please consider to use normal weight,” “overweight,” and obese” to compare the asthma outcomes instead of BMI values only. You may compare the childhood asthma risk by categorizing maternal prepregnancy BMI into different categories to see if the association is still significant. BMI categories might correlate more to clinical practice than BMI values. 

Response: We compared the childhood asthma risk by categorizing maternal prepregnancy BMI into different categories, e.g. normal weight / overweight /obese in both Tables 1 and 2.

Results

Page 4, Line 173.  56.4% + 43.7 % is not equal to 100%. Please revise the percentage numbers. 

Response: We have corrected them

Page 5 Figure 1. Please revise the resolution of your figure. It needs to be clarified for reading.

Response: We have now revided them.

Table 1. Please list the BMI information as categories (overweight n (%), obese n (%)…). 

Response: In the 2nd paragraph of our results we described BMI categories. We further performed the statistical analysis by compared BMI categories with childhood asthma and we included this information in Table 1 and Table 2.

Please explain the number of preterm birth were up to 30 %. Why? 

Response: We added in the discussion section that since the weeks of prematurity were between 35-37, we have further run an analysis separately for 35, 36 and 37 weeks;  however; we found no statistically significance difference between the weeks.

Page 6, Line 195 typos. botn?

Response: We have corrected the typos in the manuscript

Page 7, table 2. Better to compare the risk by categorized BMI groups. 

 Response: We compared the risk by categorized BMI groups, e.g. underweight & normal weight vs overweight & Obese

Discussion 

Page 9, Line 314. Citation [26-45]? Please check it. 

Response: We have now corrected it

Page 9, Line 328 citation 49

Response: We have corrected the citation

Please add more discussion about the potential mechanisms for maternal obesity, breastfeeding, and asthma development. 

Response: We have added new sections in the discussion, regarding the underlying mechanisms.

Consider doing a subgroup analysis to know if a longer duration of breastfeeding could provide better protection effects of childhood asthma. For example, please provide the information to see if children with exclusive breastfeeding duration for six months may have a lower asthma risk than children without breastfeeding. 

Response:  We further performed an analysis for a subgroup of mothers (n=2.195) with available data for exclusively breastfeading for at least 6 months without finding any significant difference compared to those bresfeeding for at least four months. We reported these findings in the results section.

Since asthma may be related to environmental factors, such as air pollution or obesity, please consider doing a subgroup analysis (younger vs. older age children; 2-3 years vs. 4-5 years) to see if the protective effect for asthma development may diminish or persist in older children. 

Response: We performed a subgroup analysis by dividing the enrolled children into two groups: children aged 2-3 years vs children aged 4-5 years, and we did not found any significant difference that may be influeced by environmental factors. We reported these findings in the results section.

Round 2

Reviewer 1 Report

Thank you for the responses. Accepted in present form 

Author Response

Thank you